# Functional Role of Single-Nucleotide Polymorphisms on IFNG and IFNGR1 in Humans with Cardiovascular Disease

**DOI:** 10.3390/ijms26188806

**Published:** 2025-09-10

**Authors:** Megh Mehta, Yang Li, Smriti Parashar, Catalina Ramirez, Heather McKay, Alan Landay, Redouane Aherrahrou, Aarushi Advani, Raag Patel, Robert Kaplan, Jason Lazar, Kathryn Anastos, David B. Hanna, Qibin Qi, Klaus Ley

**Affiliations:** 1Immunology Center of Georgia, Augusta University, Augusta, GA 30912, USA; memehta@augusta.edu (M.M.); sparashar@augusta.edu (S.P.); raherrahrou@augusta.edu (R.A.); 2Medical College of Georgia, Augusta University, Augusta, GA 30909, USA; aadvani@augusta.edu (A.A.); raapatel@augusta.edu (R.P.); 3Department of Epidemiology and Population Health, Albert Einstein College of Medicine, Bronx, New York, NY 10461, USA; yang.li@nyulangone.org (Y.L.); robert.kaplan@einsteinmed.edu (R.K.); david.hanna@einsteinmed.edu (D.B.H.); qibin.qi@einsteinmed.edu (Q.Q.); 4NYU Langone Health, Manhattan, New York, NY 11220, USA; 5Division of Infectious Diseases, UNC School of Medicine, Chapel Hill, NC 27599, USA; catalina_ramirez@med.unc.edu; 6Bloomberg School of Public Health, Johns Hopkins University, Baltimore, MD 21218, USA; hmckay4@jhu.edu; 7Department of Internal Medicine, University of Texas Medical Branch, Galveston, TX 77555, USA; allanday@utmb.edu; 8Department of Medicine, SUNY Downstate Health Sciences University, Brooklyn, New York, NY 11203, USA; jason.lazar@downstate.edu; 9Department of Medicine, Montefiore Medical Center, Bronx, New York, NY 10467, USA; kanastos@montefiore.org

**Keywords:** IFNG (interferon-gamma), IFNGR1 (interferon-gamma receptor 1), sc-eQTL (single-cell expression quantitative trait locus), T cells, SNPs (single-nucleotide polymorphisms), PBMCs (peripheral blood mononuclear cells), CVD (cardiovascular disease)

## Abstract

HIV infection is known to increase the risk for cardiovascular disease (CVD). Although almost 400 single-nucleotide polymorphisms (SNPs) are significantly associated with CAD alone, a subtype of CVD, the functions of most of these risk loci are unclear. Here, we investigated the impact of genetic variants/SNPs on the expression of nearby genes as potential cis expression quantitative trait loci (cis-eQTLs). We investigated peripheral blood mononuclear cells (PBMCs) from 31 participants in the Women’s Interagency HIV Study (WIHS) using genotyping, single-cell (sc)RNA-seq, and CITE-seq. We found 187 statistically significant sc-eQTLs (single-cell eQTLs). In total, 160 were specific for just one immune cell type. We found a set of 3 sc-eQTLs impacting expression of IFNGR1 in CD4+ T cells at the mRNA and protein level as detected by flow cytometry. Two other sc-eQTLs representing one locus impact IFNG expression in CD8+ T cells, one of the primary sources of this cytokine. The sc-eQTLs impacting IFNG were associated with Th1 (T-helper1) gene expression patterns in CD4+ T cells in this cohort. These data suggest that some individuals are genetically predisposed to greater levels of Th1 polarization, which is known to be associated with atherosclerosis.

## 1. Introduction

Most human diseases have a genetic component that influences susceptibility and prognosis. One prominent example of this is coronary artery disease (CAD), the world’s leading cause of mortality and morbidity [1]. CAD is estimated to be 40–60% heritable [2]. Another example is HIV. Although HIV infection is dependent on exposure to the virus, certain individuals are more susceptible to chronic infection [3]. Many genetic variants that influence disease risk have now been identified [4,5,6,7].

The influence of genetics on disease can be assessed by measuring the effects of common single-nucleotide polymorphisms (SNPs). Genome-wide association studies (GWASs) do this by directly correlating SNPs with disease phenotypes. However, disease status is very distal to genetics. Unmeasured transcriptional, translational, and post-translational processes can significantly influence disease progression. This makes it difficult to identify how GWAS loci may work. Even though CAD GWASs have identified nearly 400 significant genetic variants [8], the functions of many of these variants are not established. To begin to study causality and to give biological context to SNPs, expression quantitative trait locus (eQTL) mapping is useful. eQTLs are SNPs that are associated with a target gene’s expression levels. Gene expression is a more proximal phenotype that ties gene expression changes to genetics.

Unlike a GWAS, which associates SNPs directly with disease phenotypes, eQTL mapping investigates how SNPs affect gene expression in specific cells or tissues relevant to disease. For example, in humans, gene expression patterns in peripheral blood mononuclear cells (PBMCs) are strongly correlated with both CVD [9,10] and HIV [11,12], making PBMCs a particularly valuable source of cells for eQTL analysis. Unlike tissue cells, PBMCs do not need to be made into single-cell suspensions by enzymes and mechanical means. PBMCs can be cryopreserved without loss of viability for decades and are available from many biorepositories. Among PBMCs, CD4+ T cells make up the largest fraction and are very well-studied [13,14]. CD4+ T cells are known to differentiate into various subtypes based on signals in their immediate environment. These subtypes have different impacts on atherosclerosis progression. Specifically, T-helper 1 (Th1) cells are generally pro-inflammatory and atherogenic [15,16], while Th2 cell formation suppresses Th1 differentiation and is generally atherosuppressive [17]. These subtypes are also impacted by HIV infection, which causes a shift in T helper subtypes from mainly Th1 cells to mainly Th2 cells [18]. Other CD4+ T cells include Th17 [19] and follicular helper cells [20].

Most eQTL studies to date have been conducted using bulk-RNA seq of cells from individuals without disease [21]. Since bulk RNA-Seq reports the average gene expression across all cell types in a sample, eQTL mapping using this data may obscure significant correlations. This can be overcome by using single-cell RNA sequencing data (scRNA-Seq), providing single-cell transcriptomes that map single-cell eQTLs (sc-eQTLs). Sc-eQTL mapping has several advantages, particularly greater power to identify eQTLs in genes that are only expressed in a small fraction within abundant cell populations [21]. In this study, we applied an sc-eQTL mapping approach to participant PBMC samples from 31 women in the Women’s Interagency HIV Study (WIHS) [22]. These participants were in a vascular sub-study that identified whether they had subclinical cardiovascular disease (sCVD). This analysis led to the identification of 187 sc-eQTLs, two novel sets of which impact IFNG and IFNGR1 production in CD8+ and CD4+ T cells, respectively. These are furthermore associated with the expression of Th1 gene signatures.

## 2. Results

### 2.1. Study Approach

This study is based on genotyping, scRNA-Seq, and CITE-Seq data from 31 female participants in the WIHS cardiovascular sub-study. In total, 15 of them were sCVD+, defined as at least one plaque detected by standardized B mode ultrasound [23] in the right carotid artery. The median age at the time of the blood draw was 55 years. Most (>80%) had a history of smoking and recreational drug use (self-reported stimulant, non-alcoholic depressant, hallucinogenic, opioid, or cannabis) (~50%). About three-quarters of the WIHS participants (23 of 31) were HIV+ and >80% of the HIV+ women were on HAART at the time of blood draw, resulting in undetectable HIV-1 mRNA in 59% of them (Appendix A).

Targeted scRNA-Seq was conducted by BD-Rhapsody (mRNA expression levels for 485 genes) combined with CITE-Seq (40 surface markers, Appendix A from [10]). The overall workflow began with PBMC isolation, viability check, staining with the 40 oligonucleotide-tagged mAbs, followed by library preparation for gene expression (GEX) and antibodies (ADT) (Figure 1). Dimensionality reduction and clustering was achieved using the WNN algorithm [24]. The genotypes of the 31 WIHS participants were determined by Illumina Human OmniChip SNP chip analysis (2.2 million SNPs [25]) and filtered for SNPs with minor allele frequencies of at least 0.2, resulting in 705,592 SNPs considered for eQTL analysis. A total of 34,378 of these SNPs were located within 100 kbp of at least one protein-coding gene represented in the scRNA-Seq panel, and were tested for significance as sc-eQTLs (Appendix A). The number of minor alleles (0 for major allele homozygous, 1 for heterozygous, and 2 for minor allele homozygous) was significantly correlated with the expression of each of these genes for 187 SNP–gene pairs, identifying these as sc-eQTLs.

The wNN (weighted nearest-neighbors) UMAP based on 485 transcripts and 40 surface markers of 35,279 PBMCs clearly separated seven major cell types (Figure 2a): B cells, CD4+ T cells, CD8+ T cells, NK cells, and classical, intermediate, and nonclassical monocytes, identified by their classical cell surface markers CD3, CD4, CD8, CD14, CD16, CD19, and CD56 (Figure 2b).

### 2.2. Sc-eQTL Mapping and Validation

Sc-eQTLs were identified by multiple linear regression using base R and a high-performance computing environment for parallelization, filtered for SNPs that were within 100 kbp of the regulated gene. In total, 187 of these sc-eQTLs were statistically significant after *p*-value adjustment (Bonferroni), representing a significant relationship between SNP and gene in at least one cell type (Figure 3, Appendix A). Most (160 of 187) were private to one cell type (the few overlapping sc-eQTLs are shown in Appendix A).

Next, we validated the newly discovered sc-eQTLs against a published dataset in which human PBMCs were sorted by flow cytometry, bulk RNA extracted, and libraries made and sequenced [26]. Some of the sorted cell types matched the cell types in the present scRNA-Seq dataset. The directionality of the sc-eQTLs was highly correlated (Figure 4a), with all comparisons between published datasets and our datasets being 80% similar. Thus, >80% of our sc-eQTLs and the DICE eQTLs showed SNP–gene relationships that have effects going in the same direction. Furthermore, the magnitude of effect size between our sc-eQTLs and DICE eQTLs was >50% similar in all comparisons (Figure 4b).

### 2.3. IFNG and IFNGR1 Sc-eQTLs in T Lymphocytes

Prior studies have demonstrated that Th1 cell polarization, determined by TBX21 expression, is genetically based [27,28]. One significant sc-eQTL from our results impacts TBX21 expression in CD4+ T cells, exemplifying this (Appendix A). Because of the known importance of T-helper1 (Th1) polarization in many diseases including CVD [29], we focused on significant Th1-relevant sc-eQTLs. The defining Th1 cytokine is interferon-γ (gene name, IFNG, [30]). IFNG is expressed in CD8+ T cells and NK cells (Figure 5a, Appendix A). Its signaling receptor IFNGR1 is most highly expressed in monocytes and is also expressed in some CD4+ T cells (Figure 5b, Appendix A). In CD8+ T cells, we found that one sc-eQTL locus was significantly correlated with IFNG expression (Figure 5c). IFNGR1 had three significant sc-eQTLs represented by three SNPs from different loci (Figure 5d).

To test the expression of IFNGR1 at the protein level, we selected additional PBMC samples from WIHS participants expected to have high or low expression of IFNGR1, respectively, based on their genotype. Fluorescence-minus-one (FMO) was used as a negative control (Figure 6a). Expression of IFNGR1 on the CD4+ T-cell surface was verified in a HIV-/sCVD- participant. Both WIHS participants expressed IFNGR1 in some CD4+ T cells. In the high-expressing participant #26, about 19% of all CD4+ T cells expressed IFNGR1, whereas this percentage was about 13% in the low-expressing participant #10 (Figure 6a,b gating scheme in Appendix A). Thus, the sc-eQTLs for IFNGR1 resulted in a measurable difference in interferon-γ receptor at the protein level.

Based on the RNA-Seq data, CellChat identified 65 receptor–ligand interactions between the seven major cell types in PBMCs (Figure 7a, list of receptor–ligand pairs in Appendix A). Focusing on the interferon-γ signaling pathway identified CD8+ T cells and NK cells as the main sources of IFNG (Figure 7b). Interferon-γ affected all target cell types and had autocrine effects (Figure 7c). To test the biological consequences of IFNG expression in CD8 T cells, we tested whether the sc-eQTL genotype correlated with the expression of an IFNG response gene signature determined by NicheNet analysis (Appendix A). To this end, WIHS participants were stratified into IFNG high, medium, and low expressors. The high expressors had a significantly (*p* < 0.01) stronger enrichment of this signature than low expressors (Figure 7d). The master transcription factor for Th1 differentiation, T-bet (encoded by TBX21), and the receptor (IL12RB1) for another Th1 polarizing cytokine, IL-12, also had significant sc-eQTLs in CD4+ T cells (Appendix A). Taken together, these findings suggest that the sc-eQTL at IFNG in CD8+ T cells explains some degree of this Th1 polarization.

Finally, to identify if any significant sc-eQTLs from our dataset are also potential causal variants for significant GWAS loci in risk factors for atherosclerosis, we did a colocalization study. We examined whether any of our results were causal variants for CAD, lipid profile, blood pressure, or type 2 diabetes. There were no significant results from this analysis, implying that no sc-eQTL from this study alone can be considered as causing any of these clinical variables.

## 3. Discussion

Our data show that three novel sc-eQTLs significantly affect IFNGR1 mRNA and cell surface protein expression in CD4+ T cells. Additionally, two sc-eQTLs control mRNA expression in CD8+ T cells for IFNG, a Th1-polarizing cytokine [31]. The proportion of CD4+ T cells expressing IFNG-induced genes correlated with IFNG sc-eQTL status in CD8+ T cells in this cohort. These findings identify sources of common variation contributing to gene expression changes across all PBMCs, with a focus on genetic control of Th1 polarization in CD4+ T cells.

An early twin study previously investigated the genetic basis of IFN-y and T-bet expression [28]. Here, the authors identified a significant contribution of genetic background as opposed to environmental factors in determining expression of IFN-y and T-bet in CD4+ T cells. Doubtless, there are multiple locations in the genome that regulate these expressions, likely to exist in both coding and noncoding regions. Interestingly, the DICE database [26] failed to identify significant bulk eQTLs or sc-eQTLs impacting T-bet, and only identified one affecting IFN-y production in CD4+ T cells, but not CD8+ T cells. This is likely because our study investigated a cohort of primarily Black and Hispanic women, while the DICE cohort was mostly white and Asian people of both biological sexes. There are, however, indirect mechanisms that have been described. The DICE article describes, for instance, a cis-eQTL in T cells affecting GAB2, a gene whose knockdown has been shown to significantly impact IFNG levels [26].

The propensity of T→Th1 cell polarization in an individual impacts cardiovascular disease severity. There is an established bias towards cytokines that induce Th1 cells within human atherosclerotic lesions [16]. Additionally, mouse studies have shown that the proportion of Th1 cells has a direct impact on the size and number of lesions developed [32]. Our lab has identified T cells directly reactive to several atherosclerosis-specific epitopes, with a greater proportion of Th1 cells being atheroreactive than other T-cell subtypes [33]. Conversely, in HIV infection, the role of Th1 cells is unclear. It is established that individuals with HIV have more Th2 cells than Th1 cells, especially in later stages of disease [34]. However, it is unclear if this is because Th1 cells are more susceptible to the virus [35], if individuals with more Th1 cells are more resistant to becoming seropositive [36], or a combination of both.

Taken together, these studies suggest that Th1 polarization is, in part, genetically based, and Th1 cell prevalence accelerates atherosclerotic disease. Our results link these two by identifying sc-eQTLs associated with changes in Th1 cells in participants with and without sCVD. However, there are other genetic factors that influence T-cell activity. For instance, the primary transcription factor driving Th2 polarization is GATA3 [37], which was not identified as an eGene in this study. Additionally, other cell types like smooth muscle cells play a role in atherosclerotic cardiovascular disease (ASCVD) [38], and other environmental factors like diet and exercise, which were not quantified in our cohort, significantly influence atherosclerosis and Th1 polarization [39].

The present study has several limitations. There are technical limitations such as the minor allele frequency threshold and targeted gene panel that prevent this study from identifying less common sc-eQTLs that impact expression of other genes in immune cells. There are no men interrogated in the study cohort. By limiting the cohort to only women, we discovered significant sc-eQTLs that impact females, but we do not know whether they impact men. Additionally, the range of ethnicities in this cohort is narrow, and primarily consists of African American and Hispanic participants. The sample size is also too small to power discovery of all possible sc-eQTLs that may influence this population. Several studies [40,41] have shown that the effect of the additional X chromosome on gene regulation is extensive, while the differences between demographics are typically less pronounced. This implies that our results are more restricted to women than they are to African American/Hispanic populations, though this is speculation. Colocalization analysis was conducted between the full set of novel sc-eQTLs and several GWASs that investigated clinical characteristics associated with coronary artery disease, type 2 diabetes, blood pressure, BMI, and lipid panel measurements did not yield any shared causal variants. This is likely due to differences in the set of variants measured and the demographics between our study and large-scale GWASs. Future sc-eQTL studies with larger sample sizes and SNP microarrays may yield significant colocalization results.

The premier sc-eQTLs of interest were validated via flow cytometry and cell–cell communication analysis. Additionally, validation of the directionality of all our results shows good concordance with the DICE database, which we used as a reference here. Our findings primarily concern genetic control of IFNG-IFNGR1 signaling in Th1 cells. Clinical patient variables that are correlated with Th1 levels, such as coronary artery calcification and carotid intimal media thickness [17], may be more predictable given information on these variants. Sc-eQTL studies are also a new field with very few publications.

## 4. Materials and Methods

### 4.1. Participant Cohort and Sample Collection

The WIHS was established in 1993 as a multi-center, prospective, observational cohort study of women in the United States with and without HIV and other comorbidities, with clinical research sites in Atlanta, GA; Birmingham, AL/Jackson, MS; Chapel Hill, NC; Chicago, IL; Miami, FL; New York City, NY; Los Angeles, CA; San Francisco, CA; and Washington, DC [22]. Biological and behavioral data from the >4000 participants were collected every 6 months. All participants provided informed consent, and each site’s study was approved by its Institutional Review Board. In 2018, the WIHS merged with the Multicenter AIDS Cohort Study (MACS) and now continues as the MACS-WIHS Combined Cohort Study [42].

The participants from this study were involved in a vascular sub-study of 1865 participants that identified participants’ subclinical CVD status [43,44]. Participants underwent B-mode ultrasound imaging of 6 locations in the right carotid artery: the near and far walls of the common carotid artery, carotid bifurcation, and internal carotid artery. A standardized protocol was used for classification at all sites. Each image was assessed at a centralized reading center, and sCVD status was defined as the presence of one or more plaques in any of the assessed locations.

Of these 1865 participants, 92 were selected for further assessment of blood monocyte transcriptomic changes [45], which was further narrowed to 32 participants for CITE-seq analysis [10]. At the time of analysis, our primary research interest was the intersection of sCVD status and HIV positivity. As such, we selected 4 groups of 8 participants each that were quartet-matched to each other. These groups were (1) HIV−/sCVD−, (2) HIV+/sCVD−, (3) HIV+/sCVD+, and (4) HIV+/sCVD+/CRT+ (Cholesterol Reduction Therapy). HIV infection status was determined via enzyme-linked immunosorbent assay (ELISA) and confirmed via Western blot. Non-sCVD participants with self-reported coronary heart disease or current lipid lowering therapy were excluded. Participants within each quartet were matched by age at baseline visit (±5 years), visit number, smoking history, and date of specimen collection (within 1 year). Participant demographic data is summarized in Appendix A. The median age was 55 years. A total of 96% of participants self-reported as Black or Hispanic. Most reported a history of smoking, and substance use was also widespread.

### 4.2. SNP–Chip Genotyping and QC

Participant blood samples were stored at −20 °C, and a genome-wide scan was performed using the Illumina HumanOmni2.5-quad beadchip (NCBI build 38, hg38, Illumina, San Diego, CA, USA). Further details are described in [46].

Raw genotyping data were clustered and samples > 0.99 call rate were selected. This process was repeated with reclustering and filtering once again by samples > 0.99 call rate. From these steps, there resulted in zero SNPs < 0.97 call frequency, less than 0.25 AA/AB/BB R means. These filtered samples were exported as PLINK 2.0 files (Shaun Purcell/Christopher Chang, Boston, MA, USA) [47] and strand correction was performed following Will Rayner’s guide (Will Rayner, https://www.chg.ox.ac.uk/~wrayner/strand/, Accessed on 5 August 2024).

### 4.3. PBMC Preparation, Library Preparation, and CITE-Sequencing

Samples were prepared and processed as described in [10].

### 4.4. CITE-Seq and Cell Type Identification

Each participant’s PBMCs were isolated using Ficoll. The PBMC fraction from each participant’s cryopreserved blood became the starting samples for a CITE-seq experiment using the BD Rhapsody with AbSeq (BD Biosciences, Franklin Lakes, NJ, USA). The samples were processed in 8 separate plates of 4 participants each, where each plate contained a quartet-matched group of participants. These samples were processed over 2 days with at least 4 plates completed per day to minimize batch effects over multiple days.

The FASTA and FASTQ files were uploaded to Seven Bridge Genomics platform (BD Rhapsody Sequence Analysis Pipeline (Revision 17), Velsera, Boston, MA, USA), and .csv count matrices were generated for RNA and ADT data. At this point, the analysis produced 54,078 cells, with data for 496 genes and 40 antibodies. Eleven of these genes were not expressed, leaving 485 genes for analysis. This was a targeted panel focused on immune and inflammatory genes, and so sc-eQTLs impacting other genes across the transcriptome are not included in this study. Additionally, one participant’s CITE-seq data did not pass standard quality control metrics from the BD Rhapsody SevenBridges platform, and so this sample was removed from further analysis, bringing our analyzable sample n-number to *n* = 31.

### 4.5. Doublet Removal and Quality Control

Doublets were identified and removed using the DoubletFinder v2.0 package in R (Chris McGinnis, San Francisco, CA, USA) [48] resulting in the exclusion of 3225 doublets and retention of 36,321 high-confidence cells. These cells were then processed using the Seurat pipeline for further quality control and downstream analysis (Seurat v5.0, Satija Lab, New York, USA) [49]. Additional quality control filtering was performed by subsetting the Seurat object to remove cells with RNA read counts <2000 per cell, fewer than 100 detected gene features per cell, and <200 antibody-derived tag (ADT) features per cell. These criteria were applied to remove low-quality or dying cells. After these filtering steps, 35,279 cells remained for clustering and downstream expression quantitative trait loci (eQTL) analysis. One participant’s dataset was excluded due to insufficient overall transcriptomic coverage, resulting in a final cohort of *n* = 31 participants. For multimodal analysis, both mRNA and ADT data were log-normalized and CLR-normalized, respectively, scaled, and subjected to principal component analysis (PCA). Integration of these modalities was performed using the Weighted Nearest Neighbors (WNN) algorithm (https://satijalab.org/seurat/articles/weighted_nearest_neighbor_analysis, accessed on 10 April 2024), which jointly incorporates transcriptomic and proteomic data for clustering. This analysis produced a UMAP showing seven major peripheral blood mononuclear cell (PBMC) types, which were annotated based on canonical markers and used in all downstream analyses including eQTL mapping.

### 4.6. eQTL Mapping

eQTLs are classified as either cis-eQTLs or trans-eQTLs based on whether the target gene is located nearby (<100 kbp) or distally (>100 kbp) to the associated SNP, respectively [50]. cis-eQTL mapping is more feasible because it produces larger effect sizes and fewer false positives than trans-eQTLs [51,52]. Because of this, we chose to specifically perform cis-sc-eQTL mapping by integrating PLINK genotyping data with normalized CITE-seq mRNA expression. All computations were conducted on the Einstein College of Medicine High-Performance Computing Cluster. The original genotyping data, with a total of 2,221,034 variants, were filtered to retain only variants with a minor allele frequency (MAF) > 0.2 to ensure adequate allelic representation across the cohort. This led to a final total of 705,592 analyzable variants. This was a necessary step in the study to maximize statistical power, but as a result, our findings do not identify sc-eQTLs from less common variants. Discovering those sc-eQTLs will require larger sample sizes. mRNA expression data were pseudobulked by participant to generate an average gene expression matrix for 485 genes from each of the seven annotated cell types. This pseudobulk matrix was used as the response variable in a multivariable linear regression model, with genotype as the primary explanatory variable. The following covariates were included in the model to control for confounding: age, smoking status, ethnicity, the first three principal components (PCs) of the genotyping data, and disease group status. This also limits the effect of variables related to disease group, like HAART status, on eQTLs. To limit the number of tests and improve statistical power, the analysis was restricted to SNPs located within 100 kilobase pairs (kbp) of a gene’s transcription start site. This threshold was based in part on findings from [53], which reported that most functional enhancer–gene interactions occur within 100 kbp in K562 cells. The statistical power of this study is further increased by a high minor allele frequency threshold of 0.2 or greater. Given *n* = 31 individuals, a Bonferroni threshold of 0.05 for 35,279 tests, and MAF ≥ 0.2, the study achieves ~50% power to detect effect sizes of ~1.53 SD per allele, ~80% power at ~1.80 SD, and ~90% power at ~1.94 SD. eQTLs with smaller effects are underpowered in this sample size. The regression produced beta coefficients and standard errors for each tested SNP–gene pair. After Bonferroni correction for multiple testing, 187 SNP–gene associations met the significance threshold and were classified as eQTLs. Of these, 13 lacked widely used rsIDs, despite having phase SNP IDs from the 1000 Genomes Project. The remaining 175 eQTLs had identifiable rsIDs and were used for all subsequent analyses.

### 4.7. Replication/Validation Analysis

To determine whether our set of sc-eQTLs was consistent with previously published findings, we compared our results to those from the Database of Immune Cell Expression (DICE) (https://dice-database.org/, Vijayanand Lab, San Diego, CA, USA, accessed on 23 February 2025) [26]. The DICE repository contains bulk and single-cell eQTL data, available as unfiltered and filtered variant call format (.vcf) files. We downloaded the unfiltered datasets for Naïve B cells, Naïve CD4+ T cells, Naïve CD8+ T cells, Monocytes, M2 macrophages, and NK cells from the DICE database.

We extracted effect sizes (betas) and calculated standard errors from these datasets, combining them with corresponding metrics from our own sc-eQTL analyses. To ensure relevant comparisons, we retained only eQTL results located within 500 kbp of significant sc-eQTLs identified in our dataset (Appendix A). These combined datasets of effect sizes and standard errors were used as inputs for the mashR v0.2.79 analysis (Stephens Lab, Chicago, IL, USA) [54]. mashR employs a Bayesian approach to estimate the posterior probability of concordance between our results and those from the DICE database. For stringent concordance analyses we applied a factor threshold of 0.5.

To test whether the proportion of validated eQTLs depended on the genomic distance threshold, we additionally repeated mashR analyses using alternative genomic distances of 100 kbp, 250 kbp, and 1 Mbp. The results did not differ significantly across these thresholds, indicating stable concordance between our findings and those from the DICE database regardless of genomic proximity criteria.

### 4.8. Flow Cytometry

Cryopreserved PBMC samples from 2 of the original 31 WIHS participants and controls were thawed at 37 °C in a water bath and washed with PBS (without Ca/Mg) by centrifugation at 400× *g*, 10 min. Cells were counted using a hemocytometer, and viability was determined using the Trypan Blue dye (Thermofisher, # 15250061, Boston, MA, USA) exclusion method. Cells were washed again with cold flow cytometry buffer [PBS w/o Ca/Mg, 2% fetal bovine serum (FBS)] at 400× *g*, 5 min, 4 °C. Subsequently, 1–2 million cells were stained with antibodies and reagents listed in Appendix A for 40 min, 4 °C. Cells were finally washed, resuspended in FACS buffer, and data was acquired on Cytek Aurora 5-Laser Spectral Cytometer from the AU Flow and Mass Cytometry Core. Voltages were set up using single-color-stained cells and compensation beads (Invitrogen, # 01-2222-42, Boston, MA, USA). Data was analyzed using FlowJo version 10.10.0 (BD Bioscences, Franklin Lakes, NJ, USA).

### 4.9. NicheNet and CellChat Analyses

All analyses were conducted using R v4.3.3 (R Core Team, NZ) [55] and RStudio v2024.12.1+563 (Posit, MA, USA) [56].

Using the Seurat object generated from post-CITE-sequencing data analysis, the NicheNet v2.2.0 (Saeys Lab, Belgium) [57] and CellChat v2.1.2 (Nie Lab, San Francisco, CA, USA) [58] packages were together capable of determining which ligand–receptor interactions are most significant in a target cell population, which cell types are capable of sending those ligands, and what genes are affected upon ligand reception.

CellChat analysis was performed with tutorial-provided parameters for each pipeline function, except for the interaction_input file and the computeCommunProb() function. The interaction_input file was modified to include an interaction between IFNG and IFNGR1 alone as an interaction term, as our dataset did not contain information on IFNGR2, which was required in the provided interaction_input file. The computeCommunProb() function is responsible for computing whether an interaction is occurring based on the number of ligand–receptor interactions that are counted from previous steps in the pipeline. Here, we utilize the “truncatedMean” option with a trim = 0.05 (the fraction of results to exclude from either end of the analysis). This provides leads to the pipeline providing a greater number of interactions that are overall weaker compared to the default “triMean” method, which has a trim = 0.25.

NicheNet analysis was performed with tutorial-provided parameters for each pipeline function. The defined condition of interest was the “HIV+/sCVD+” subgroup of participants, while the reference condition was the “HIV+/sCVD−” group of participants. The minimum expression percentage threshold was 0.05. The sender cell group was identified CD8+ T cells/CD4+ T cells/B cells (negative control)/NK cells. The receiver cell group was identified as CD4+ T cells. The top 3 ligand–receptor interactions identified by AUPR score were considered significant. The active target threshold for downstream ligand–receptor interaction effects was 500 genes downstream of the initial interaction—this is a large number of interactions, yielding 104 total significant interactions. A tight threshold of regulatory targets >25th percentile in strength was drawn. In all, this meant that only the top 25% of these genes were considered significantly impacted by ligand–receptor interaction.

### 4.10. Colocalization

All analyses were conducted using R v4.3.3 (R Core Team, NZ) [55] and RStudio v2024.12.1+563 (Posit, MA, USA) [56]. Colocalization analyses were conducted using the HyPrColoc v1.0 (Foley Lab, Cambridge, UK) algorithm described here [59]. This algorithm takes multiple clinical phenotypes as endpoints to identify causal factors via a Bayesian approach to the probability that a sc-eQTL from a given dataset is likely to be the causal variant behind any GWAS loci from another dataset. To identify GWAS datasets to query our results against, we searched SNPs from our assay in the t2d.hugamp [60] online resource for GWAS that are related to heart disease and overall health, focusing on type 2 diabetes, blood pressure, BMI, lipid profile, and CAD. Colocalization GWAS datasets that were queried are cited here: [61,62,63,64,65]. Each GWAS input dataset was filtered for effect size and standard error matrices that were merged with sc-eQTL effect size and standard error matrices to determine if sc-eQTLs could be causing GWAS loci hits.

## 5. Conclusions

We discovered and validated 2 sets of sc-eQTLs that control IFNγ expression in CD8+ T cells and IFNGR1 expression in CD4+ T cells. These genes are a sender–recipient pair, and the sc-eQTLs regulating IFNγ expression are associated with Th1 polarization of CD4+ T cells.

## Figures and Tables

**Figure 1 ijms-26-08806-f001:**
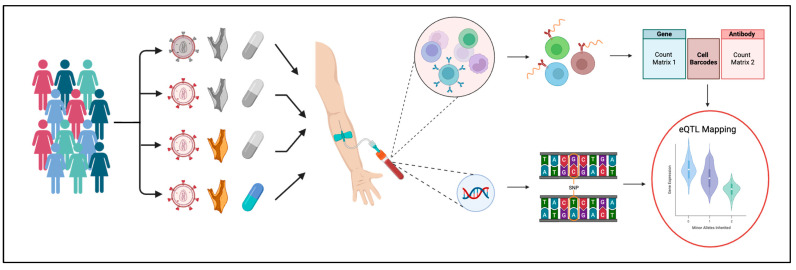
**Participants, sample preparation, and analysis pipeline.** WIHS participants were classified into groups of HIV (virus symbol)−/sCVD (carotid symbol)−, HIV+/sCVD−, HIV+/sCVD+, and HIV+/sCVD+/LLT (lipid-lowering therapy, pill symbol)+. Blood was taken from participants for CITE-seq (top) and SNP microarray (bottom). These data together enable eQTL mapping (red circle) Created in BioRender. Mehta, M. (2025) https://BioRender.com/ax2sr7r, accessed on 5 August 2025.

**Figure 2 ijms-26-08806-f002:**
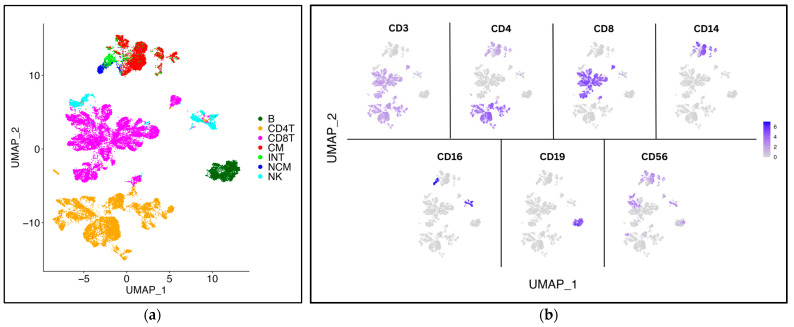
**WNN (weighted nearest-neighbors) UMAP of PBMCs from 31 WIHS participants.** (**a**) UMAP (SLM clustering, resolution = 0.2), INT = intermediate monocytes, NCM = non-classical monocytes, CM = classical monocytes; (**b**) main surface marker expression based on CLR-normalized, thresholded ADT signals from CITE-Seq (ADT values on scale).

**Figure 3 ijms-26-08806-f003:**
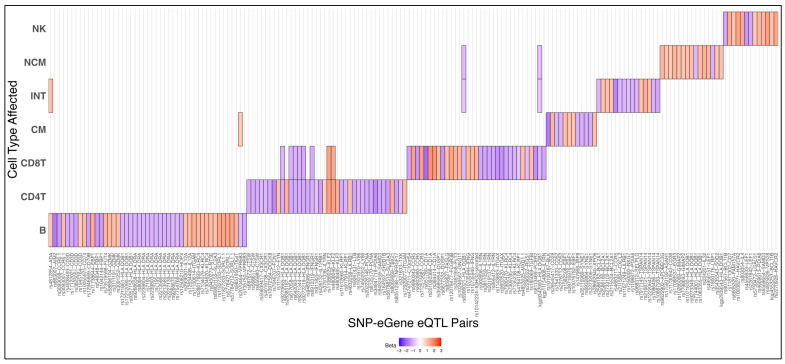
**Significant eQTLs are largely cell type specific:** All 187 statistically significant (Bonferroni FDR < 0.05) eQTLs were sorted by FDR value from lowest (red) to highest (blue) and plotted for each cell type. A total of 160 eQTLs were significant in only one cell type, 9 in two cell types, and 3 in more than two cell types. eQTLs significant in multiple cell types detailed in Appendix A.

**Figure 4 ijms-26-08806-f004:**
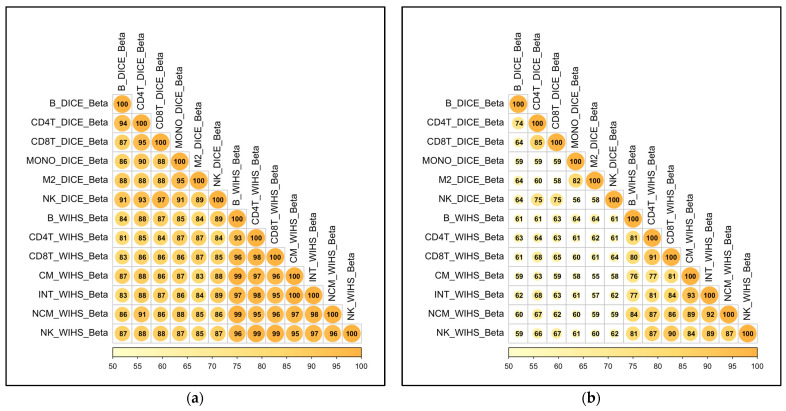
**eQTL validation against the DICE dataset (FACS-sorted human PBMCs).** (**a**) Percentage of eQTLs whose effect size directionality replicated between our dataset and the DICE database for different PBMC types (filtered for eQTLs found within 500 kbp of a significant eQTL [shown in Figure 3] in our dataset); (**b**) percentage of eQTLs whose effect size and directionality replicated between the present dataset and the DICE database, where replicates are effect sizes within a factor of 0.5 of each other.

**Figure 5 ijms-26-08806-f005:**
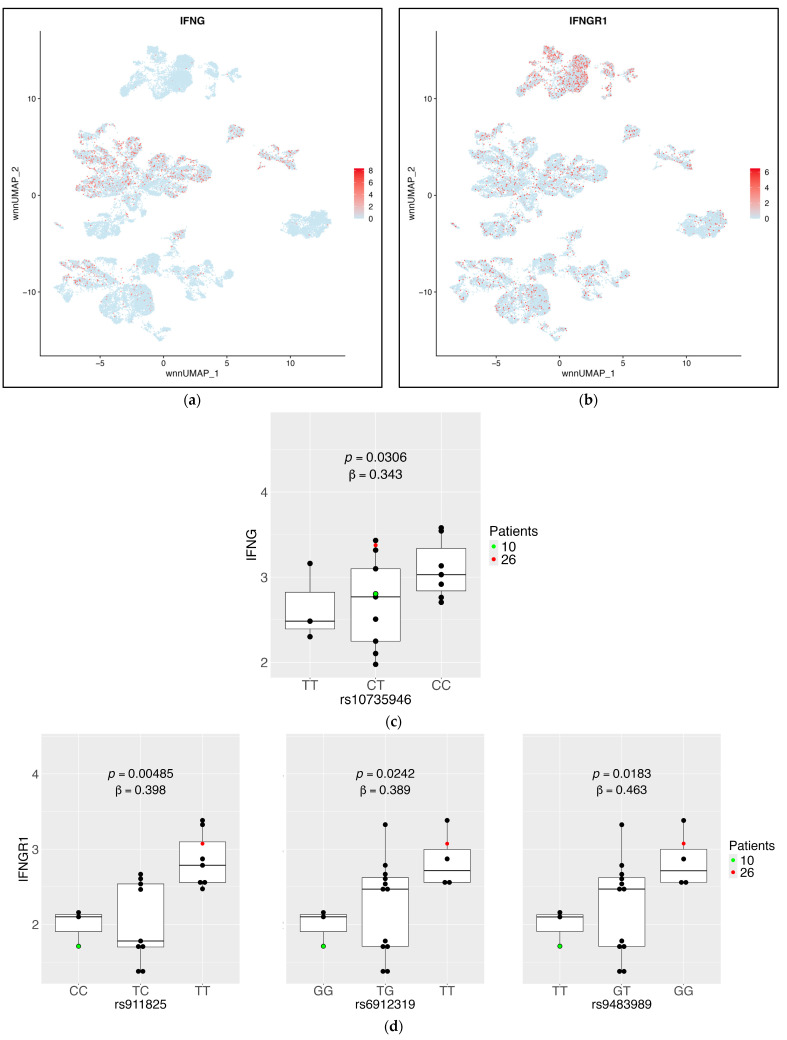
**eQTLs in T cells significantly impact expression of IFNG and its receptor IFNGR1.** (**a**) Log-normalized mRNA expression of IFNG in all cells, CD8+ T cells highlighted; (**b**) log-normalized mRNA expression of IFNGR1 in all PBMCs, CD4+ T cells highlighted; (**c**) log-normalized mRNA expression of IFNG in CD8+ T cells per participant as a function of sc-eQTL genotype at one locus. Each dot is one WIHS participant; the box indicates median, quartiles, and range. Participants 10 and 26 (highlighted) were used for flow cytometry validation; (**d**) log-normalized mRNA expression of IFNGR1 in CD4+ T cells per participant as a function of sc-eQTL genotype at 3 distinct loci, details as in (**c**).

**Figure 6 ijms-26-08806-f006:**
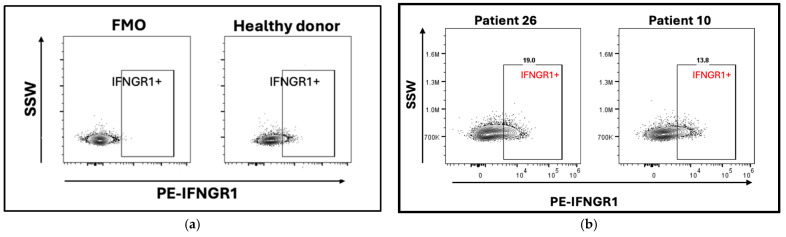
**IFNGR1 cell surface protein expression by flow cytometry.** Gated on CD4+ T cells (see Appendix A). (**a**) Fluorescence-minus-one (FMO) negative control and healthy non-WIHS donor positive control; (**b**) IFNGR1 expression in WIHS participant 26 (high expressor from Figure 4: rs911825 TT, rs6912319 TT, and rs9483989 GG) and WIHS participant 10 (low expressor: rs911825 CC, rs6912319 GG, and rs9483989 TT).

**Figure 7 ijms-26-08806-f007:**
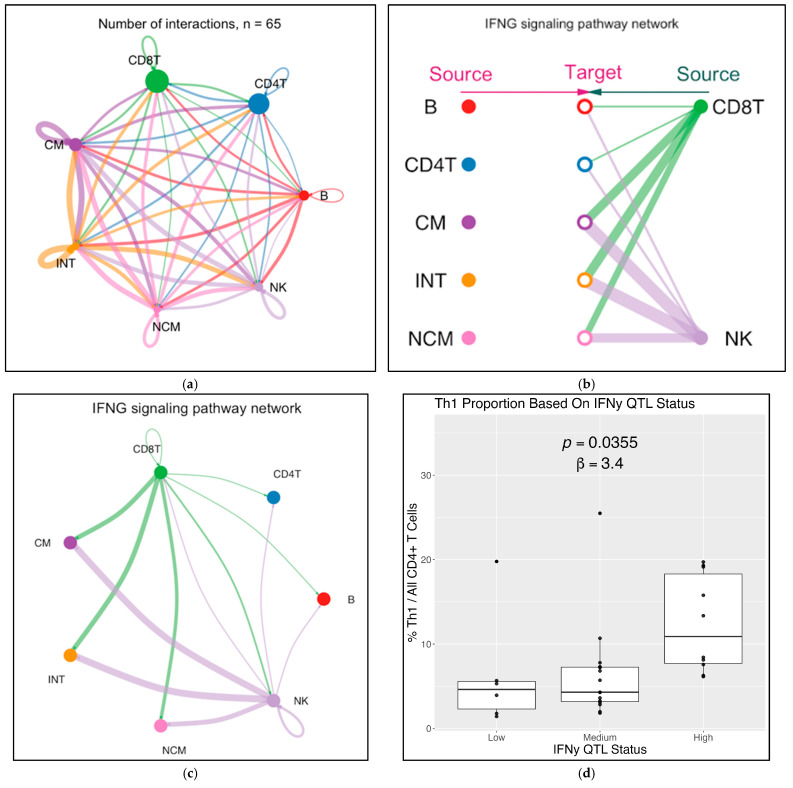
**Cell interaction analysis by CellChat.** (**a**) All cell–cell interactions between 7 major cell types in all WIHS participant PBMCs. Color indicates cell of origin, thickness indicates number of interactions. Receptor–ligand pairs in Appendix A; (**b**) CD8+ T cells and NK cells as major sources of IFNG; (**c**) IFNG signaling network; (**d**) CD4+ T-cell co-expression of IFNG-regulated genes is predicted by IFNG sc-eQTL status in CD8+ T cells. Based on sc-eQTL status at these loci, WIHS participants were classified as high, medium, or low IFNG expressors. List of genes used for co-expression score in Appendix A.

## Data Availability

All post-analysis data can be found in figures and Appendix A. Raw FASTQ data on the CITE-seq experiment is available on the Gene Expression Omnibus at GSE205320. Genotyping data for the participants in this study is restricted for privacy reasons, and any access to this data requires approval from the MWCCS (MACS/WIHS) combined study center at the NIH.

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
