# Peer review of "Functional Role of Single-Nucleotide Polymorphisms on IFNG and IFNGR1 in Humans with Cardiovascular Disease"

_ijms, 2025, doi:10.3390/ijms26188806_

Round 1
Reviewer 1 Report
Comments and Suggestions for Authors
Dear Authors
The manuscript reports single-cell cis-eQTL (sc-eQTL) mapping in PBMCs from 31 women who were enrolled in the cardiovascular sub-study of the WIHS, most of whom were Black or Hispanic and living with HIV. The authors identify 187 significant sc-eQTLs, 160 of which are cell-type specific, using SNP genotyping, CITE-seq (40 surface markers), and targeted scRNA-seq (485 genes). They draw attention to (i) a single locus containing two sc-eQTLs linked to CD8+ T cell IFNG expression and (ii) three sc-eQTLs at different loci linked to CD4+ T cell IFNGR1 expression. They offer two participants flow cytometry-based limited protein-level validation for IFNGR1, and they use CellChat/NicheNet analyses to infer functional implications that point to a genetically influenced Th1 polarization axis. Shared causal variants are not produced by colocalization with GWAS traits. In order to tune Th1 biology related to atherosclerosis, the study suggests a genetically driven IFNG–IFNGR1 sender–receiver axis.
Advantages:
- integrating genomic and transcriptome data at the single-cell level in a well-studied cohort.
- Verification through flow cytometry and cross-validation with an outside resource (DICE).
- Using sophisticated cell communication modeling tools, a biologically significant pathway (IFN-γ) is analyzed.
Major revisions:
- Sample size and generalizability: There are no men in the cohort; it is limited to 31 women, most of whom are African American and Hispanic. Please:
- Provide evidence for the statistical power to identify strong sc-eQTL effects.
- Extend the conversation about the potential impact of this demographic restriction on generalizability.
- Technical limitations: Despite being mentioned in passing, it is important to specifically address the potential biases brought about by SNP filtering (MAF > 0.2) and the use of a preselected 485-gene scRNA-seq panel.
- Consider discussing which associations could have been missed.
- Causality and functional analysis: Additional investigation is necessary due to the absence of colocalization with CAD or associated GWAS loci:
- provide potential justifications (differences in the population, definition of phenotypes, statistical power).
- Make recommendations for upcoming functional validation techniques (e.g., larger cohorts, cellular models).
- Translational implications Clarify the potential clinical relevance: could these findings inform risk stratification or therapeutic targeting?
Minor revisions:
- Correct small typographical/formatting errors in tables and figures.
- Make sure all acronyms, such as sc-eQTL and PBMC, are defined when they are used for the first time in the main text.
- Increase the readability of the figures (Figures 3 should have more detailed axis labels, for example).
- Check references for consistent IJMS formatting.
Author Response
Thank you for taking the time to review this article and provide comments. We appreciate the feedback you have provided to improve this manuscript. Please see below the full list of comments and responses/revisions we have made to the revised manuscript, where all corresponding changes are highlighted in the re-submitted files.
Comments 1: 1. Sample size and generalizability: There are no men in the cohort; it is limited to 31 women, most of whom are African American and Hispanic. Please: Provide evidence for the statistical power to identify strong sceQTL effects. Extend the conversation about the potential impact of this demographic restriction on generalizability.
Response 1: We agree that the sample size and generalizability of this study is limited to women and African American/Hispanic populations. The statistical power is maximized as much as possible by a high MAF threshold, but this has other technical issues mentioned in Comments 2. Therefore, we have extended the discussion to elaborate on how the set of results is restricted to sex and demographics to differing degrees in pg.9, paragraph 3, lines 259-263, as well as discussed statistical power in more detail in the methods section in pg. 12, paragraph 1, lines 381-385. We extend the discussion with references to articles discussing sex and human demographics as biological variables in research.
Comments 2: Technical limitations: Despite being mentioned in passing, it is important to specifically address the potential biases brought about by SNP filtering (MAF > 0.2) and the use of a preselected 485-gene scRNA-seq panel. Consider discussing which associations could have been missed.
Response 2: We agree that there are some technical limitations that lead to sc-eQTLs which exist in this cohort but are missed by the study prior to mapping. To address this, we have edited pg. 9, paragraph 3, lines 252-254 to reflect that these limitations prevent the study from identifying less common sc-eQTLs in gene sets outside of the targeted panel. We have also modified our explanation of the targeted panel in the methods section on pg. 11, paragraph 1, lines 330-332.
Comments 3: Causality and functional analysis: Additional investigation is necessary due to the absence of colocalization with CAD or associated GWAS loci: provide potential justifications (differences in the population, definition of phenotypes, statistical power). Make recommendations for upcoming functional validation techniques (e.g., larger cohorts, cellular models).
Response 3: We agree that there are potential reasons as to why colocalization analysis did not yield any common variants which could be discussed further. To address this, we have modified the discussion section, pg. 9, paragraph 3, lines 267-269, which details why our variants may not have overlapped with large-scale GWAS. This also includes a recommendation that larger studies with larger SNP microarray panels may be more capable of finding such overlaps.
Comments 4: Translational implications Clarify the potential clinical relevance: could these findings inform risk stratification or therapeutic targeting?
Response 4: Agreed. To address this concern, we have expanded on the discussion of clinical applications of this research in the final paragraph of the discussion section on pg. 9, paragraph 4, lines 272-276. We discuss research that has identified clinical characteristics associated with Th1 polarization that may benefit from genetics-based prediction using SNP data from this study.
Minor Comments 1: Correct small typographical/formatting errors in tables and figures.
Minor Comments 3: Increase the readability of the figures (Figures 3 should have more detailed axis labels, for example).
Minor Response 1/3: We edited the readability of 3 of the main figures so that the formatting is more uniform and the figures more understandable. Specifically, axes labels have been added to Figure 3, the naming convention of the rows and columns has been standardized in Figure 4, the size of the text font in Figure 7 has been standardized, and the abbreviations table on pg. 15 has been fixed.
Minor Comments 2: Make sure all acronyms, such as sc-eQTL and PBMC, are defined when they are used for the first time in the main text.
Minor Response 2: All acronyms used are now defined at their first instance. A definition of sc-eQTL is included in the abstract as the first reference of sc-eQTL, in pg.1, paragraph 1, line 30. PBMC is defined in the abstract and its first mention in the introduction.
Minor Comments 4: Check references for consistent IJMS formatting.
Minor Response 4: Agreed. The reference style has been updated to reflect the modern IJMS standards for references and citations, spanning pages 15-20.
Reviewer 2 Report
Comments and Suggestions for Authors
Review of the manuscript “Significant impact of single nucleotide polymorphisms on IFNG and IFNGR1 in humans with cardiovascular disease” for the IJMS (ISSN 1422-0067) journal.
The manuscript investigates the impact of single nucleotide polymorphisms (SNPs) on IFNG and IFNGR1 expression in CD8+ and CD4+ T cells from women with cardiovascular disease (CVD) and HIV infection. The authors employed SNP genotyping, single-cell RNA sequencing (scRNA-Seq), and CITE-Seq to identify 187 sc-eQTLs, including loci specifically affecting IFNG and IFNGR1 expression. The findings suggest that certain individuals have a genetic predisposition for Th1 polarization, which is linked to atherosclerosis.
The topic is timely and relevant for understanding genetic mechanisms in cardiovascular disease. The use of scRNA-Seq and CITE-Seq allows for single-cell resolution and more accurate eQTL mapping. Results are logically connected to immune function and Th1 polarization. Validation with flow cytometry and comparison to the DICE database enhances reliability.
During the review of this manuscript though, some remarks and comments appeared.
Minor comments:
- The cohort includes only women, predominantly of African American and Hispanic ancestry, limiting generalizability.
- The sample size is small (n=31), reducing statistical power and potentially missing other sc-eQTLs.
- No colocalization with GWAS loci for cardiovascular risk factors was observed.
After addressing these minor comments, the manuscript would be suitable for publication.
Comments on the Quality of English LanguageThe manuscript is generally written in clear and scientifically appropriate English. Terminology is used consistently, and grammatical structures are mostly correct. The descriptions of methods and results are logical and understandable. Some sentences are very long and complex, which can affect readability; breaking them into shorter sentences would improve clarity.
Author Response
Thank you for taking the time to review this article and provide comments. We appreciate the feedback you have provided to improve this manuscript. Please see below the full list of comments and responses/revisions we have made to the revised manuscript, where all corresponding changes are highlighted in the re-submitted files.
Comments 1: The cohort includes only women, predominantly of African American and Hispanic ancestry, limiting generalizability.
Response 1: Agreed. This was also a concern brought up by other reviewers. Therefore, we have included modifications to the discussion section at pg. 9., paragraph 3, lines 259-263, lengthening the discussion of how the study demographics limits the application of the results.
Comments 2: The sample size is small (n=31), reducing statistical power and potentially missing other sc-eQTLs.
Response 2: Agreed. We have changed the methods section subtitled eQTL mapping as well as points in the discussion section to reflect this technical limitation and detail this issue explicitly to the reader. Because of the small sample size, we limited our study to eQTLs with large minor allele frequencies (>0.2). These changes can be found on pg. 12, paragraph 1, lines 381-385.
Comments 3: No colocalization with GWAS loci for cardiovascular risk factors was observed
Response 3: To elaborate on this, we have changed the discussion section and changed our interpretation of the results section of colocalization. These can be found at pg. 9, paragraph 3, lines 267-269 and pg. 8, paragraph 1, lines 210-211.
Comments 4: The manuscript is generally written in clear and scientificallyappropriate English. Terminology is used consistently, andgrammatical structures are mostly correct. The descriptions ofmethods and results are logical and understandable. Somesentences are very long and complex, which can affect readability;breaking them into shorter sentences would improve clarity.
Response 4: Agreed. We have made some minor edits to parts of the introduction where sentences were longer than necessary. These changes can be found on pg. 2, paragraph 1, line 52; pg. 2, paragraph 2, line 66; pg. 2, paragraph 3, line 83.
Reviewer 3 Report
Comments and Suggestions for Authors
Dear Authors,
I have reviewed your article entitled “Significant impact of single nucleotide polymorphisms on IFNG and IFNGR1 in humans with cardiovascular disease.” I would like to congratulate you, as the presentation is both innovative and well-structured. However, I have identified some areas for improvement:
-
I believe the title is somewhat too technical. You might consider simplifying it to make it more appealing, using terms such as Functional role or Genetic regulation of.
-
In the abstract, there are some vague expressions (lines 35–36), for instance: “some individuals have a genetic propensity for Th1 polarization.”
-
Throughout the text, the phrase “for example” is used repeatedly; it could be beneficial to use alternatives.
-
In lines 207–211, referring to colocalization with GWAS, you state that “there were no significant results.” I suggest reinforcing this statement with a more detailed interpretation from the authors.
In addition, I would kindly ask the authors to address the following questions and consider incorporating the responses into the main text:
-
Is the sample size sufficient to detect robust eQTLs, given the large genetic variability?
-
How might antiretroviral therapy affect gene expression and the identified eQTLs?
-
You mention that no significant results were found when cross-referencing with GWAS of CAD, lipids, blood pressure, and diabetes. Could this reflect limitations in statistical power or differences in ethnicity/cohort compared to GWAS populations (which are mostly European)?
Finally, the conclusion may be somewhat too direct and assertive. It does not take into account that Th1 polarization also depends on environmental factors (nutrition, infections, microbiota...). I recommend rephrasing it as follows: “Our findings suggest that genetic variants could contribute, together with environmental factors, to the predisposition toward Th1 polarization.”
Author Response
Thank you for taking the time to review this article and provide comments. We appreciate the feedback you have provided to improve this manuscript. Please see below the full list of comments and responses/revisions we have made to the revised manuscript, where all corresponding changes are highlighted in the re-submitted files.
Comments 1: I believe the title is somewhat too technical. You might consider simplifying it to make it more appealing, using terms such as Functional role or Genetic regulation of.
Response 1: Agreed. We have changed the title to ‘Functional role of single nucleotide polymorphisms…’ on pg. 1 to reflect this change, as we believe this is a better title than the original, more vague ‘significant impact.’
Comments 2: In the abstract, there are some vague expressions (lines 35– 36), for instance: “some individuals have a genetic propensity for Th1 polarization.”
Response 2: Agreed. This was changed on pg. 1, paragraph 1, line 36, to describe that some individuals are predisposed to higher levels of Th1 cells, rather than the vaguer original form of that sentence.
Comments 3: Throughout the text, the phrase “for example” is used repeatedly; it could be beneficial to use alternatives.
Response 3: We have now replaced one of the occurrences on pg. 2, paragraph 3, line 70, with ‘specifically’, as it better describes that particular example’s relationship to the axiom it exemplifies.
Comments 4: . In lines 207–211, referring to colocalization with GWAS, you state that “there were no significant results.” I suggest reinforcing this statement with a more detailed interpretation from the authors.
Response 4: Agreed. We have changed the final sentence of the results section here with a more detailed impression of those results on pg. 8, paragraph 1, lines 210-211, specifically that they imply that no sc-eQTL from this study alone causes those clinical variables.
Question 1: Is the sample size sufficient to detect robust eQTLs, given the large genetic variability?
Q-Response 1: Agreed. This concern was brought up by other reviewers as well, and we have addressed this in the discussion section and the methods sections in pg. 9, paragraph 3, lines 258-259, pg.9, paragraph 3, lines 268-269, and pg. 11, paragraph 3, lines 368-370. These sections provide recommendations for larger sample sizes in the future and discuss the implications of the reduced statistical power of this study on the results.
Question 2: How might antiretroviral therapy affect gene expression and the identified eQTLs?
Q-Response 2: Agreed. Antiretroviral therapy will affect gene expression, but our study is not focused on eQTLs that may be affected by this. We further describe regressing out disease group status as a covariate in the linear regression model and how this relates to HAART status on pg. 11, paragraph 3, lines 374-377. Because of these covariates, by definition, these eQTLs describe a relationship between genotype and gene expression with the effect of disease state regressed out.
Question 3: You mention that no significant results were found when crossreferencing with GWAS of CAD, lipids, blood pressure, and diabetes. Could this reflect limitations in statistical power or differences in ethnicity/cohort compared to GWAS populations (which are mostly European)?
Q-Response 3: We agree with this critique. We discuss limitations of the populations and statistical power on the colocalization datsets chosen to do our comparison with here, with the modifications on pg. 9, paragraph 3, lines 267-269. These should make this possibility clear to the reader.
Question 4: Finally, the conclusion may be somewhat too direct and assertive. It does not take into account that Th1 polarization also depends on environmental factors (nutrition, infections, microbiota...). I recommend rephrasing it as follows: “Our findings suggest that genetic variants could contribute, together with environmental factors, to the predisposition toward Th1 polarization.”
Q-Response 4: To highlight that Th1 polarization is also subject to other factors, such as environmental, we have included mention of Th1 levels explicitly in pg.9, paragraph 2, line 251, showing that other factors not measured in this study impact these levels.
Round 2
Reviewer 1 Report
Comments and Suggestions for Authors
Subsequent to a comprehensive examination of the manuscript, "Significant impact of single nucleotide polymorphisms on IFNG and IFNGR1 in humans with cardiovascular disease," I am delighted to announce the authors have effectively addressed the minor concerns articulated during the review period. The alterations provided meet the necessary standards and resonate well with our journals editorial requirements. Further modifications isn't really needed currently.